# The effect of gut passage by waterbirds on the seed coat and pericarp of diaspores lacking "external flesh": Evidence for widespread adaptation to endozoochory in angiosperms

Mihai Costea[1]*, Hiba El Miari[1], Levente Laczkó[2], Réka Fekete[2], Attila V. Molnár[2], Ádám Lovas-Kiss[3°], Andy J. Green[4°]

1 Department of Biology, Wilfrid Laurier University, Waterloo, Ontario, Canada, 2 Department of Botany, University of Debrecen, Debrecen, Hungary, 3 Department of Tisza Research, Wetland Ecology Research Group, MTA Centre for Ecological Research-DRI, Debrecen, Hungary, 4 Wetland Ecology Department, Estación Biológica de Doñana (EBD-CSIC), Sevilla, Spain

° These authors contributed equally to this work.
* mcostea@wlu.ca

**Data Availability Statement:** All relevant data are available in the Supporting Information files.

## Abstract

The widely accepted "endozoochory syndrome" is assigned to angiosperm diaspores with a fleshy, attractive tissue and implies the existence of adaptations for protection against digestion during gut passage. This syndrome has led diaspore fleshiness to be emphasized as the exclusive indicator of endozoochory in much of the ecology and biogeography research. Crucially, however, endozoochory in nature is not limited to frugivory, and diaspores without "external flesh" are commonly dispersed, often over long distances, via birds and mammals by granivory. A key question is: are such diaspores somehow less prepared from an architectural point of view to survive gut passage than fleshy diaspores? To answer this question, we selected 11 European angiosperm taxa that fall outside the classical endozoochory syndrome yet are known to be dispersed via endozoochory. We studied their seed coat/pericarp morphology and anatomy both before and after gut passage through granivorous waterfowl, and determined their seed survival and germinability. We found no fundamental differences in the mechanical architecture of the seed coat and pericarp between these plants dispersed by granivory and others dispersed by frugivory. Neither diaspore traits *per se*, nor dormancy type, were strong predictors of diaspore survival or degree of damage during gut passage through granivores, or of the influence of gut passage on germinability. Among our 11 taxa, survival of gut passage is enabled by the thick cuticle of the exotesta or epicarp; one or several lignified cell layers; and diverse combinations of other architectural elements. These protection structures are ubiquitous in angiosperms, and likely to have evolved in gymnosperms. Hence, many angiosperm diaspores, dry or fleshy, may be pre-adapted to endozoochory, but with differing degrees of specialization and adaptation to dispersal mechanisms such as frugivory and granivory. Our findings underline the broad ecological importance of "non-classical endozoochory" of diaspores that lack "external flesh".

**Funding:** MC lab was funded by NSERC Discovery Canada (327013). AMV was supported by National Research, Development and Innovation Office (K108992), and AJG by Spanish Ministerio de Economía, Industria y Competitividad project CGL2016-76067-P (AEI/FEDER, EU). ÁL-K was supported by a János Bolyai Research Scholarship of the Hungarian Academy of Sciences (2019-2021) and by the New National Excellence Program of the Ministry of Human Capacities (ÚNKP-19-4-DE-172). The funders had no role in study design, data collection and analysis, decision to publish, or preparation of the manuscript.

**Competing interests:** The authors have declared that no competing interests exist.

# Introduction

*"This list (of plants germinated from the excreta of birds) embraces such a diversity of seed types . . . that one may reasonably suppose that most seeds and fruits within a certain range of size may at one time or another be dispersed by this agency in a viable condition"*

(Salisbury 1961)

Endozoochory—the dispersal of diaspores or propagules after ingestion and passage through a digestive system—is one of the earliest documented mutualisms involving plants and animals [1]. More recently, multiple authors argued that the morphology of diaspores largely reflects their dispersal agent (e.g., [2–3]) and such putative morphological adaptations of diaspores have been classified into "dispersal syndromes" [4], which have become very popular in the literature and are commonly assumed to represent functional traits. The most important criteria of van der Pijl [4] for the recognition of diaspores with an "endozoochory syndrome" were: "1) an attractive edible (outer) part and 2) an inner protection of the seed against digestion". Further differences in colour, size, odour, position and display of diaspores on the plant, and water/nutrient content have been linked to the feeding limitations or preferences of the various frugivorous organisms acting as dispersal vectors, including birds, mammals and reptiles (e.g., [4–6]). The evolution of plant traits favoring endozoochory has been seen as a "diffuse" co-evolutionary process (e.g., [7–9]).

Morphological dispersal syndromes (including also anemochory, hydrochory and epizoochory) have gained wide usage in the literature, yet assigning syndromes to an entire flora depends on a human observer making major assumptions about dispersal mechanisms largely in the absence of field observations. Syndromes have also drawn criticism because: a) diaspores may be dispersed by multiple agents (both biotic and abiotic) at the same or different stages of the dispersal process (polychory; [e.g. 10–11]; b) dispersal syndromes do not describe, qualitatively or quantitatively, disperser effectiveness (sensu Schupp [12]; e.g., [13]); and c) dispersal syndromes may limit theoretical ecological or evolutionary generalizations (e.g., [13]).

Another limitation of dispersal syndromes is that diaspores of many plants do not fit into any of the major syndromes, in which case their dispersal has been called "unassisted" [14, 15], "barochory" [16] or "unspecialized" [17]. For the purpose of our study, we consider these three terms as synonyms. "Unspecialized" diaspores are relatively small seeds released from dehiscent fruits or of dry, indehiscent fruits [2, 14], and represent the *majority* of European angiosperms [14]. Seeds/fruits of such plants are assumed to merely fall from the mother plant by gravity [15]. However, plants with "unspecialized" diaspores represent a large proportion of oceanic island floras such as those of the Galapagos (47% of endemics and 50% of non-endemic natives; [18]; Azores (63%, i.e. exactly the same as continental Europe; [17]) and the Canary Islands (78.7% natives; [19]), which implies long distance dispersal (LDD) and raises questions about the predictor value of dispersal syndromes for LDD [20–21]. Species with "unspecialized" diaspores are also dominant in the Arctic [22] and alpine habitats [23–24], and many such species have ample, circumboreal distributions and genetic patterns that imply LDD events inconsistent with syndromes [25]. The general solution to such a paradox has been to invoke "nonstandard means of dispersal", i.e. the ability of plants to disperse by (unknown) means, not predicted by their syndrome [20].

To assume that current dispersal syndromes reflect actual mechanisms of dispersal is to ignore the reality that endozoochory of plants without a fleshy fruit ("non-classical endozoochory"; [26]) is widespread in nature. The best studied examples come from granivorous and herbivorous waterbirds [27–30], and herbivorous mammals [e.g., 31–33]. Granivory enables a

seed dispersal mutualism because of constraints on food processing that prevent destruction of all ingested seeds [27–28]. The capacity of granivorous, migratory waterbirds for LDD events is greater than that of other vectors, and current dispersal syndromes have no value for predicting these events [29, 34].

Our overall objective in this study is to answer the following key question: are diaspores lacking fleshy tissues dispersed by granivorous birds somehow less prepared to survive gut passage than fleshy diaspores dispersed by frugivores? A significant body of literature has accumulated about the effect of endozoochory on seed germination, but this has mainly focused on frugivory [reviewed by 35–37]. The thickness of the "seed coat", understood in a broad sense to also include the pericarp or other embryo covering structures, was deemed important for the survival of various ingested diaspores by different frugivorous vertebrates [38], and research on granivorous waterbirds also suggests that seed coat thickness is important [39]. However, the defense role played by specific tissues within the seed coat or the pericarp during gut passage has received little attention [40–41]. Nevertheless, given the considerable evidence to suggest that endozoochory is a major dispersal mechanism for many plants not assigned to an "endozoochory syndrome" (including those considered to have "unspecialized" diaspores), the significance of diaspore protective layers for these plants needs to be investigated in more detail.

To answer the question above, we conducted a study to demonstrate the potential of a broad selection of angiosperms for endozoochory by granivorous waterfowl. Specific objectives were to: 1) study the morphology and anatomy of the seed coat/pericarp before and after waterfowl gut passage in 11 species of European angiosperms that lack an endozoochory syndrome, yet are known or likely to be dispersed by waterfowl endozoochory; 2) to determine their seed survival and germinability and relate them to morphology, anatomy and dormancy type, to assess whether these diaspores possess structural traits that are potentially adaptations for endozoochory; 3) consider if there are essential differences between diaspores within a fleshy-fruit and other diaspores in terms of seed morphology and the defenses they have to survive gut passage; 4) discuss the evolution of different types of diaspores; 5) reassess the value of the current endozoochory syndrome, which excludes non-fleshy diaspores.

## Materials and methods

### 1. Sampling and feeding assays

Eleven central European species without an endozoochory syndrome were selected (Table 1) using multiple criteria: diaspores of congenerics are known to be dispersed through endozoochory by granivorous or herbivorous waterbirds [30, 40, 42]; they represent a broad phylogenetic sampling across angiosperms (including monocots and eudicots from various plant families); they are either seeds or dry, indehiscent fruits; they are very diverse morphologically (e.g., size, shape); they originate from unitegmic or bitegmic ovules; have dissimilar types of dormancy (physical or physiological, *sensu* [36]); have varied habitat requirements in terms of soil moisture to reflect the broad range recorded for waterbird endozoochory [29, 42]. Of the 11 species, eight have physiological dormancy (PD) and three have physical dormancy (PY, see Table 1; data based on [36]). Borhidi WB moisture values [43] range from 4 to 10 (Table 1). Although only the *Elatine* spp. and *Bolboschoenus planiculmis* with a Borhidi WB value of 10 or more can be considered as aquatic plants, waterbirds disperse many terrestrial plants by endozoochory [29, 42].

We force-fed captive mallards (*Anas platyrynchos*) with a mixture of diaspores, comprised of a known number of propagules of each plant species. The diaspores were all collected during late summer from different habitats in Hungary (Table 1). No field site access permit was

**Table 1. Species analysed; diaspore type; embryological data (An = anatropous ovule; Cam = campylotropous ovule; He = hemianatropous ovule; bi = bitegmic ovule; uni = unitegmic ovule); dormancy (PD = Physiological dormancy; PY = Physical Dormancy); Borhidi WB value (moisture indicator, 1 –plants of extremely dry habitats or bare rocks, 12- water plants, entirely submersed in water; [43]); location and habitat of collection sites.**

| Plant family | Species | Diaspore type | Ovules | Dormancy type | Borhidi WB | Location of collection sites | Habitat at collection site |
|---|---|---|---|---|---|---|---|
| **Amaryllidaceae** | *Allium angulosum* | Seed | An, bi | PD | 8 | Tiszaújváros, 47.95 N, 21.07 E | Mesotrophic wet meadow |
| **Cyperaceae** | *Cyperus flavescens* | Fruit (Achene) | An, bi | PD | 8 | Monostorpályi, 47.41 N, 21.77 E | Drying ditch |
| | *Bolboschoenus planiculmis* | Fruit (Achene) | An, bi | PD | 10* | Karcag, 47.24 N, 20.86 E | Rice field |
| **Poaceae** | *Echinochloa crus-galli* | Fruit (Caryopsis) | He, bi | PD | 7 | Lakitelek, 46.87 N, 20.05 E | Inundated arable field |
| **Asteraceae** | *Cirsium brachycephalum* | Fruit (Cypsela) | An, uni | PD | 9 | Monostorpályi, 47.38 N, 21.76 E | Salt meadow |
| **Caryophyllaceae** | *Lychnis coronaria* | Seed | Cam, bi | PD | 4 | Monostorpályi, 47.41 N, 21.78 E | Poplar plantation |
| **Convolvulaceae** | *Cuscuta lupuliformis* | Seed | An, uni | PY | 8 | Tiszaújváros, 47.95 N, 21.07 E | Riverine willow-poplar forest |
| **Elatinaceae** | *Elatine hungarica* | Seed | An, bi | PD | 10 | Pocsaj, 47.28 N, 21.84 E | Inundated arable field |
| | *Elatine hydropiper* | Seed | An, bi | PD | 10 | Tiszagyenda, 47.37 N, 20.54 E | Oligotrophic lake |
| **Fabaceae** | *Astragalus contortuplicatus* | Seed | Cam, bi | PY | 8 | Tiszaroff, 47.39 N, 20.39 E | Floodplain alluvium |
| | *Glycyrrhiza echinata* | Seed | Cam, bi | PY | 7 | Lakitelek, 46.87 N, 20.05 E | River dyke |

required because seeds were harvested from unprotected, public areas. However, *Elatine hungarica*, *E. hydropiper*, *Astragalus contortuplicatus*, *Cirsium brachycephalum* and *Lychnis coronaria* are protected species in Hungary and were sampled with the permission of the Hortobágy National Park Directorate (Permit numbers.: 45-2/2000, 250-2/2001).

On each feeding trial, each mallard received 100 seeds each of ten taxa (*Allium angulosum*, *Astragalus contortuplicatus*, *Bolboschoenus planiculmis*, *Cirsium brachycephalum*, *Cuscuta lupuliformis*, *Cyperus flavescens*, *Echinochloa crus-galli*, *Elatine hungarica*, *Elatine hydropiper*, *Lychnis coronaria*) and 50 propagules each of the larger-seeded *Glycyrrhiza echinata*. Until the beginning of the experiment, the diaspores were dry-stored in paper bags in the fridge at 4˚C. Seeds used in control germination tests were stored under identical conditions. Nine mallard individuals were used for the experiment and, in order to increase sample size, each individual was used in three feeding trials. Prior to the experiments and in between feeding trials, mallards were housed communally in outdoor facilities and were fed with mixed grains (maize, wheat, oats) and green leaves (e.g., *Stellaria media*, *Taraxacum officinale*). Grit was freely available to the birds outside the experimental trials. Twenty four hours prior to force-feeding, ducks were moved to individual cages (50 × 50 × 50 cm) and kept without food, to ensure their digestive tracts were relatively empty and to minimise potentially confounding effects of other food items in the digestive tract. Water was provided ad libitum throughout the study. Individual cages were built of wire mesh and a clean plastic sheet was placed under each cage once force-feeding was completed, to allow controlled faecal sample collection. The force-feeding was done using a small plastic cone placed in the bird's throat. All diaspores were gently poured into the oesophagus of the birds.

Droppings were collected from the sheets placed under the cages at five time points following force-feeding, at 4, 7, 21, 31- and 45-hours post-feeding. After 45 h, the experiments were ended, and the ducks were returned to a communal pen. Faecal samples were left to dry at room temperature, then the intact diaspores were collected and counted in each sample under

a stereo microscope. We considered diaspores to be "intact" when they were not chipped or broken. The percentage of "intact" diaspores provided the overall seed survival.

Mallards were obtained from a local breeder and were one year-old at the time of the experiment. They were hand-reared to reduce stress in close proximity to people and showed no signs of ill effects after the experiment, and were returned to the local breeder. The experiment was approved by the scientific council of the Babeş-Bolyai University of Cluj Napoca (reference number: 14689/31.08.2018).

## 2. Germination tests

After seed separation, the surface of half of the passed diaspores was sterilized for one minute with 25% sodium hypochlorite (NaOCL) wash in a laminar air flow cab, to avoid fungal and bacterial infections. They were then washed with sterile deionized water three times for three minutes to remove remaining NaOCL. Fifty diaspores per species were used as controls and treated as above. The viability of the diaspores was tested with germination tests run for 195 days in Petri-dishes filled with 1% sterilised agarose gel. We used 14h light (30 μmol m-2 sec-1 light intensity) and 10h dark photoperiod with 22 ± 2°C daytime and 18 ± 2°C night-time temperature.

## 3. Morphology and anatomy of diaspores

The anatomy of the seed coat or pericarp had not previously been studied in these 11 species. One hundred control and 30 passed "intact" diaspores were analyzed for morphology and seed coat or pericarp anatomy. All the seeds were rehydrated in warm water for two to three days and first examined morphologically under a Nikon SMZ800 stereomicroscope to assess major morphological changes after digestion: modifications in color, cracking of the seed coat or pericarp, etc. Subsequently, diaspores were cross-sectioned manually by inserting them into one-year old stem fragments of *Hibiscus moscheutos* (comprised mostly of soft wood), a method that we found to be more rapid and causing less artifacts than the use of a cryostat. Sections were stained with Toluidine Blue O (TBO) (pH 4.4) to differentiate lignified versus cellulosic cell walls, and examined under a Nikon Eclipse 50i brightfield. At this stage, some of the sections were also stained with Tetrazolium 1% to determine if their embryo tissues were alive. Optical microscopy imaging and measurements were conducted with a PaxCam Arc digital camera and Pax-it 12 software (MIS Inc., Villa Park, IL).

The remaining diaspores, plus the sections described above, were dehydrated through a series of ethanol and processed with a Tousimis Autosamdri 931critical dry point instrument. Samples were coated with 30 nanometers of gold using an Emitech K 550 sputter coater. Examination, measurements and pictures were taken at 10 kV using a Hitachi SU1510 variable pressure scanning electron microscope (SEM). Examined diaspores were preserved as vouchers in the Wilfrid Laurier University Herbarium.

## 4. Statistical analyses

Putative relationships were explored between several morphological/anatomical traits of control diaspores and five progressive degrees of damage defined for the passed diaspores after their examination (see Results). Some of these traits (e.g. load and water permeability) were measured for a previous study (see Table 2; [44]). Then, these continuous and categorical variables were correlated with diaspore survival and germinability. We conducted Pearson correlations for continuous variables, and non-parametric Spearman correlations for damage categories. Diaspore traits were also compared between dormancy types (Table 1) using t-tests,

**Table 2. Diaspore traits: Diaspore size, cuticle thickness, total thickness of embryo covering structures (seed coat, pericarp or fertile lemma and palea for *Echinochloa*), and total thickness of mechanical layers.** Data on seed survival, wet load and seed permeability from [44].

| Species | Diaspore size (mm) | | Cuticle thickness (µm) | Thickness of embryo covering structures (µm) | Thickness mechanical layer(s) (µm) | Seed survival | Wet load (Kg) | Permeability |
|---|---|---|---|---|---|---|---|---|
| | Length | Width | | | | | | |
| **Monocots** | | | | | | | | |
| *Allium angulosum* | 1.88 | 1.22 | 0.63 | 22.27 | 12.52 | 11.7 | 2.391 | 1.186 |
| *Bolboschoenus planiculmis* | 2.81 | 2.57 | 4.12 | 168.47 | 91.29 | 48.7 | 6.615 | 1.155 |
| *Cyperus flavescens* | 0.97 | 0.69 | 0.67 | 28.79 | 26.21 | 24.6 | 0.168 | 1.178 |
| *Echinochloa crus-galli* | 3.16 | 1.98 | 0.49 | 122.41 | 93.86 | 6.75 | 1.264 | 1.19 |
| **Eudicots** | | | | | | | | |
| *Cirsium brachycephalum* | 2.86 | 1.04 | 3.84 | 111.35 | 46.83 | 6.31 | 0.471 | 0.087 |
| *Lychnis coronaria* | 0.93 | 0.98 | 2.53 | 29.81 | 21.25 | 4.44 | 0.731 | 0.019 |
| *Cuscuta lupuliformis* | 2.75 | 1.93 | 0.57 | 118.82 | 61.33 | 14.4 | 13.92 | 0.025 |
| *Elatine hungarica* | 0.611 | 0.391 | 0.12 | 29.6 | 24.3 | 21.8 | 0.052 | 0.059 |
| *Elatine hydropiper* | 0.528 | 0.228 | 0.18 | 25.85 | 22.93 | 9.45 | 0.05 | 0.059 |
| *Astragalus contortuplicatus* | 0.96 | 0.71 | 1.33 | 39.18 | 21.63 | 51.0 | 1.695 | 0.05 |
| *Glycyrrhiza echinata* | 3.96 | 2.67 | 2.29 | 108.33 | 72.42 | 7.91 | 11.713 | 0.034 |

and Levine's test to confirm homogeneity of variances. All analyses were conducted with Statistica 13.0 (Dell Inc., 2015)

## Results

### 1. Seed coat and pericarp architecture before ingestion; layers that provide protection against the mechanical and chemical effects of digestion

Morphology and anatomy of the seed coat or pericarp were compared for each taxon between control and passed diaspores (see S1 Appendix for details; Figs 1–5). The most important quantitative architectural diaspore traits are summarized in Table 2. A glossary of anatomical terms has been prepared (S2 Appendix).

As expected considering their broad taxonomic range, the control diaspores of the 11 species exhibited a broad array of morphologies and structural features. Seeds varied in length from 0.5 mm in *Elatine* spp. (Fig 2J) to 5.5 mm in *Glycyrrhiza echinata* (Fig 2R). Shape was also diverse: reniform to subround in the seeds that originated from campylotropus ovules (the two Fabaceae, *Astragalus contortuplicatus* and *G. echinata*; Fig 2N and 2R) to ellipsoid, obovoid or even cylindrical-curved in the species with anatropous ovules (*Allium angulosum*, *Cuscuta lupuliformis* and *Elatine* species: *E. hungarica* and *E. hydropiper*; Figs 1A, 2E and 2J). Epidermal sculptures were characteristic to each species, from nearly smooth or smooth (*Bolboschoenus*, Fig 3I and 3J; *Cyperus*, Fig 3N and 3O; *Cirsium* Fig 4A and 4B and Fabaceae, Fig 5J and 5R) to reticulate (*Allium angulosum*, Fig 3A and 3B; *Elatine* spp.; Fig 5A and 5F) or tuberculate (*Lychnis coronaria*, Fig 4E and 4F); with patterned epicuticular wax deposits in the latter species (Fig 4G) and in Fabaceae (Fig 5L and 5S). Species with prominent sculpture retain feces on their surface after gut passage (Fig 2C and 2L).

All the seeds examined had one main resistance cell layer represented by cells with variously lignified cells walls (S1 Appendix). This mechanical cell layer consisted of either the seed epidermis (exotesta; *A. angulosum*, Figs 1B, 3C and 3D; *A. contortuplicatus*, Figs 2O and 5M; *G.*

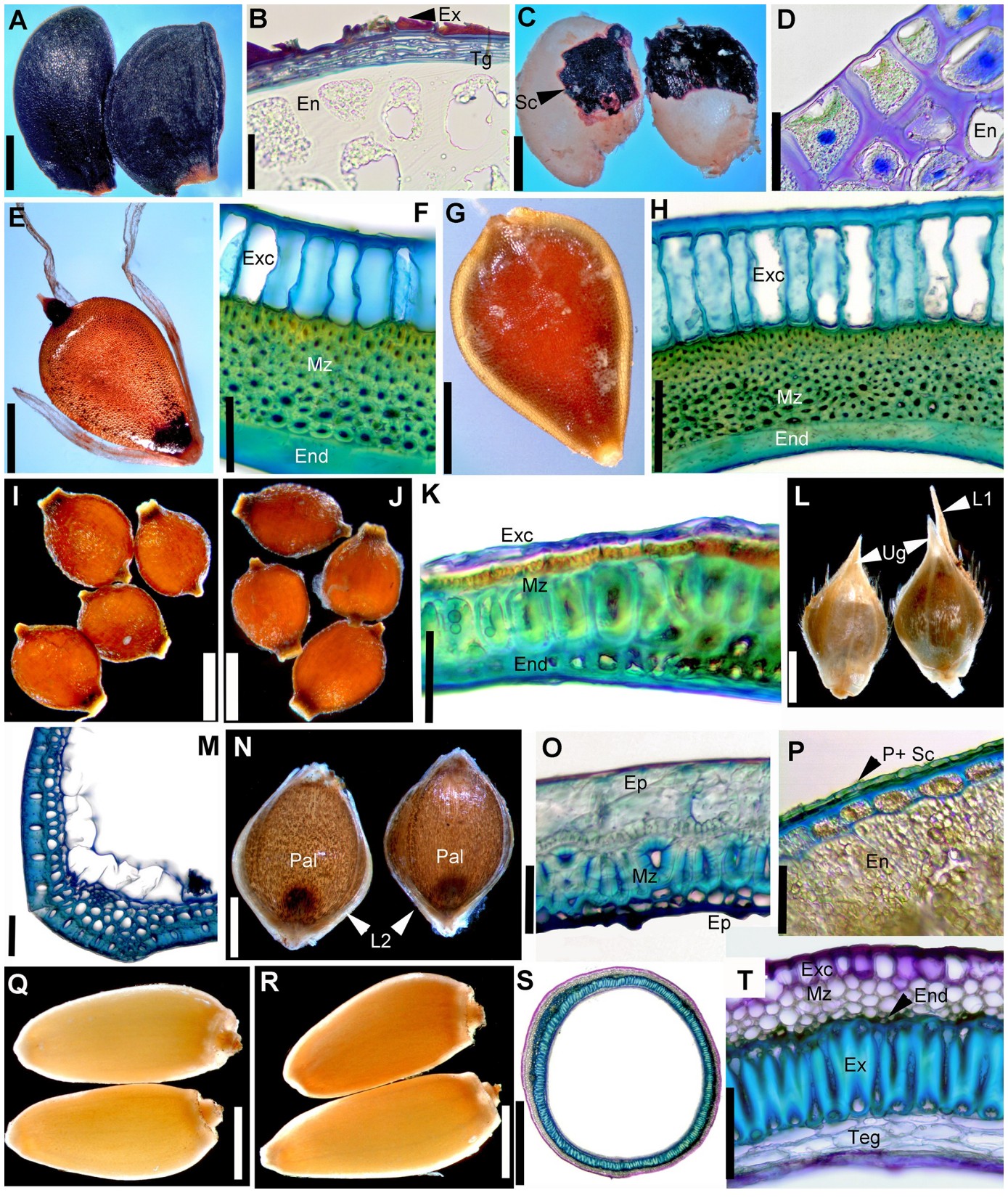

**Fig 1. Morphology of control/passed diaspores and the histology of the seed coat or pericarp.** A–D. *Allium angulosum*. A–B. Control seeds. A. General morphology. B. Anatomy of seed coat. C–D. Passed seeds. C. Overall morphology; note that the seed coat was entirely removed from large portions of the seeds. D. Endosperm left at the surface of seed (cross-section). E–H. *Bolboschoenus planiculmis*. E–F. Control achenes. E. Morphology. F. Cross-section through pericarp. G–H. Passed achenes. G. Morphology. H. Pericarp anatomy in unaffected area. I–K. *Cyperus flavescens*. I. Control achenes. J–K. Passed achenes. J. Morphology. K. Cross section through pericarp in undamaged area. L–P. *Echinochloa crus-galli*. L. Control spikelets. M. Anatomy of upper glume from control spikelets. N–P. Passed diaspores. N. Morphology; note that although the upper glume and sterile lemma were removed, the caryopses are intact within the fertile lemma and palea. O. Anatomy of fertile lemma. P. Cross-section through caryopsis after removal of fertile lemma and palea. Q–T. *Cirsium brachycephalum*. Q. Control cypselae. R–S. Passed cypselae. R. Morphology. S. Cross-section through pericarp (embryo extracted at sectioning). T. Anatomy of pericarp and seed coat. En = endosperm; End = endocarp; Ep = epidermis; Ex = exotesta; Exc = exocarp; L1 = sterile (awned) lemma; L2 = fertile lemma; Mz = mesocarp; Ms = mesophyll; Pal = palea; P = pericarp; Sc = seed coat; Teg = tegmen; Ug = upper glume. Scale bars. A, C, E, G, L, N = 1 mm; B, D, K, P = 30 μm; F, H, M, O, T = 50 μm; I, J, Q, R, S = 0.5 mm.

*echinata*, Fig 2S and *L. coronaria*, Figs 2B and 4H), a hypodermal cell layer (*C. lupuliformis*, Figs 2F and 4P) or endotesta (*Elatine* spp., Figs 2K, 5B and 5G).

Dry, indehiscent fruit diaspores included the caryopsis of *Echinochloa crus-galli* (Fig 1N, Poaceae), the achenes of *Cyperus flavescens* (Figs 1I and 3N) and *Bolboschoenus planiculmis* (Figs 1E and 3I; Cyperaceae), as well as the cypsela of *Cirsium brachycephalum* (Figs 1Q and 4A; Asteraceae). The caryopsis of *Echinochloa* is enclosed and dispersed together with the spikelet parts among which the fertile flower developed (Figs 1L and 3S). These spikelet parts are herbaceous (see S1 Appendix) and easily removed. In contrast, the fertile lemma and palea are smooth and strongly indurated around the caryopsis at maturity (Figs 1N and 3U). The external epidermis of the fertile lemma and palea is so strongly lignified that their cell lumen becomes completely obturated, and these two spikelet parts obviously act as armour during gut passage (Figs 1O and 3V). Caryopsis pericarp is very thin, represented by 1(–2) cellulosic layers of cells, fused with the seed coat, which is reduced to one single layer of cells followed by the aleurone layer of the endosperm (Figs 1P and 3V). Thus, in the case of *Echinochloa* diaspores, the pericarp and seed coat provide no protection.

The achenes of *Bolboschoenus* and *Cyperus* resemble each other somewhat in their morphology; most notably in their color, shape and nearly smooth surfaces (Fig 1E and 1I, but see S1 Appendix), as well as a similar seed coat structure consisting of only two layers of small cells with thin, cellulosic walls. However, in *Bolboschoenus* the exocarp (pericarp epidermis) cells are large, radially elongated and function as an aerenchyma (Figs 1F and 3K), while in *Cyperus* epidermal cells are small (Fig 3P). Also, the cuticle of exocarp cells in *Bolboschoenus* is thick (2.4–5 μm), while in *Cyperus* it is thin (0.4–0.8 μm).The mesocarp is reinforced by multiple layers of sclerenchyma fibers in both species: quite minimalistic in *Cyperus*, with only 2–3 cell layers (Fig 3P), and more elaborate in *Bolboschoenus* with 6–8 layers of fibers (Figs 1F and 3K). At least one 90° change in the orientation of the fibers was observed in both species.

The *Cirsium* cypsela (called achene by some authors, e.g., Roth [45]), is oblong-obovate, smooth (Figs 1Q and 4A); the pericarp consists of a cellulosic exocarp with thick cuticle (2–4.2 μm), a well-developed, parenchymatic mesocarp and lignified endocarp followed by the seed coat. However, the main mechanical layer is the exotesta represented by one layer of sclereids with peculiar lignin thickenings (Fig 1T; S1 Appendix).

Tannins in various cell layers were noted in *Allium*, *Cuscuta*, *Elatine*, *Lychnis*, *Cyperus* and *Bolboschoenus* (S1 Appendix).

In summary, the protective layers can have different ontogeny and position within the seed coat or pericarp architecture. They can originate from the ovule integuments as parts of the seed coat, from the ovary walls as elements of the fruit pericarp, or even from structures external to the flower (*Echinochloa*). The seed diaspores examined had one main protective layer of lignin cells, while the dry, indehiscent fruits exhibited multiple such layers. Protective layers are positioned either at the exterior of the diaspores (exotesta for seeds or the exocarp for fruit

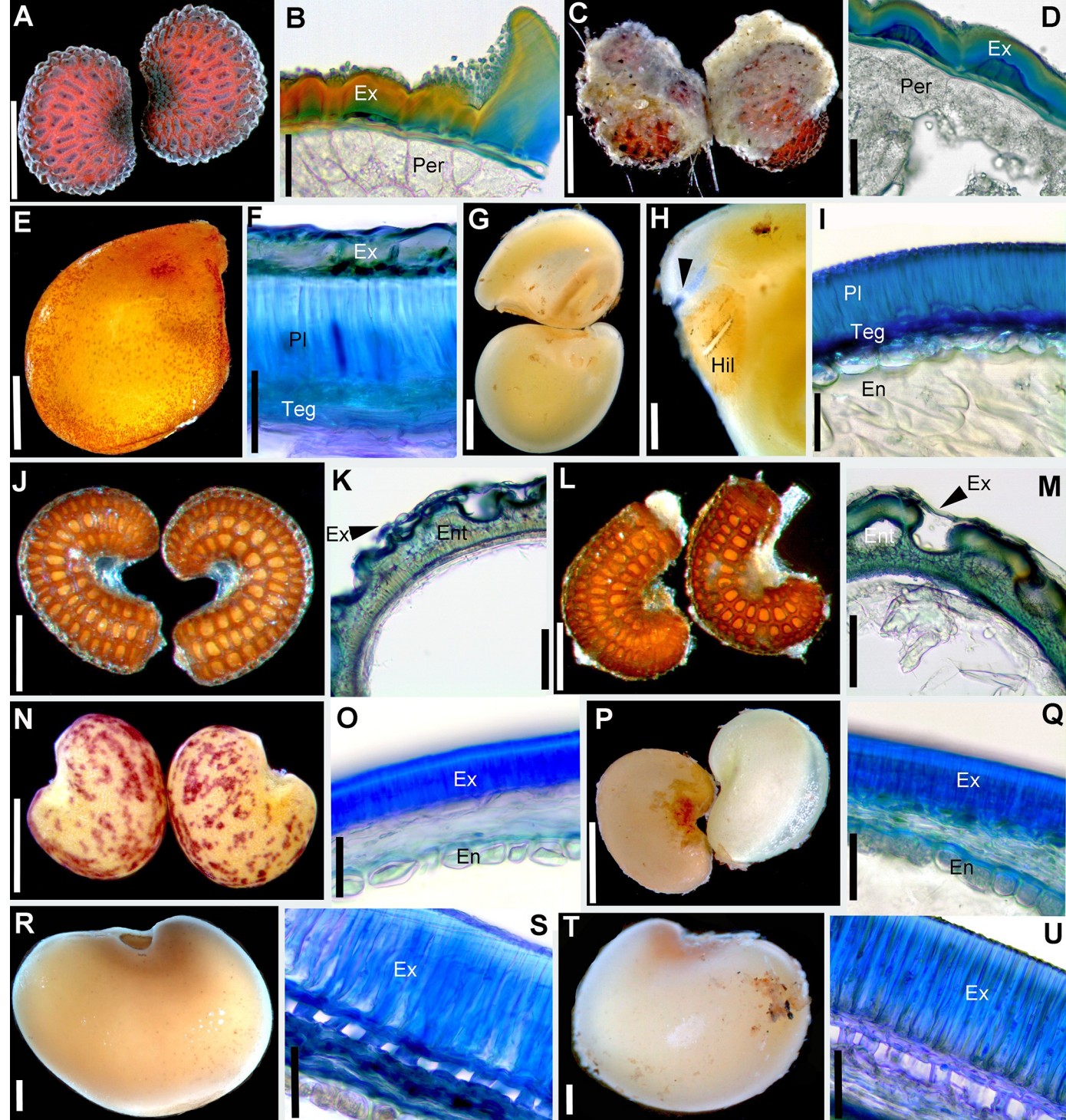

**Fig 2. Morphology of control/passed diaspores and the histology of the seed coat.** A–D. *Lychnis coronaria*. A–B. Control seeds. A. General morphology. B. Cross-section through the seed coat. C–D. Passed seeds. C. Morphology; note abundant feces adhering to the seed coat. D. Anatomy of seed coat; note removed epicuticular wax compared to control seeds. E–I. *Cuscuta lupuliformis*. E–F. Control seeds. E. General morphology. F. Anatomy of seed coat. G–I. Passed seeds. G. Morphology. H. Large cracks (arrow) opened above the hilum area. I. Cross section through seed coat and endosperm showing the palisade layer left as the most exterior seed layer. J–M. *Elatine hungarica*. J–K. Control seeds. J. Morphology. K. Cross-section through the seed coat. L–M. Passed seeds are virtually identical to control seeds. L. General morphology; note feces adhering to pitted endotesta. M. Cross-section through seed coat. N–Q. *Astragalus contortuplicatus*. N–O. Control seeds. N. Seed morphology. O. Cross-section through seed coat and endosperm. P–Q. Passed seeds. P. General morphology. Q. Cross-section through seed coat. R–U. *Glycyrrhiza echinata*. R–S. Control seeds. R. General morphology. S. Anatomy of seed coat. T–U. Passed seeds. T. General morphology. U. Anatomy of

seed coat. En = endosperm; Ex = exotesta; Per = perisperm; Pl = palisade layer; Teg = tegmen. Scale bars A, C, H, N, P, R, T = 0.5 mm; B, D, K, M, O, Q = 30 μm; E, G = 1 mm; I, F, S, U = 50 μm; J, L = 250 μm.

diaspores) or more internally—in our species, hypodermis and endotesta for seeds, and the mesocarp and/or endocarp for fruits.

## 2. Modifications of the seed coat and pericarp caused by passing through the digestive system

Overall, passed diaspores displayed a remarkable range of morphological and structural modifications: from virtually unscathed in some species, to multiple cell layers removed in others. Also, while in some species diaspores were more or less uniformly affected by passage, others exhibited a range of morphological and structural damage as a result of digestion (see below, S1 Appendix and Figs 1–5). However, as a rule, regardless of the extent of damage observed in the seeds recovered, passing did not affect the integrity of the embryo in any of the species.

The position of the mechanical layer(s) usually indicates which tissues will be affected by passing. Thus, an epidermis position (exotesta or exocarp) commonly contains the damage caused by digestion at the periphery of the diaspores, while a deeper location within the seed coat or pericarp signals that the more external tissues will be affected. This was largely true for *Astragalus* (Figs 2P–2Q and 5N–5Q), *Glychirrhiza* (Fig 2T and 2U), *Lychnis* (exotestal seed coat architecture; Figs 2D and 4J–4L), *Cuscuta* (mechanical layer in hypodermis, Figs 2G–2I and 4Q–4T), *Bolboschoenus* and *Cyperus* (mesocarp and endocarp lignified; Figs 1G, 1H, 3L, 3M; 1J, 1K, 3Q and 3R, respectively). However, in the remaining species the position of protective layer(s) was a poor predictor of the extent of damage caused by digestion. The least affected diaspores in this study were the cypselae of *Cirsium*, which exhibited only minor abrasions of their exocarp cuticle after passing (Figs 1R–1T, 4C and 4D). In this species, the tight, rectangular exocarp cells endowed with a thick cuticle were sufficient to provide protection during gut passage (the exocarp does not possess lignin reinforcements). Likewise, the endotestal seeds of *Elatine* displayed similar patterns of epidermis collapse (*E. hungarica*; Figs 2M and 5C–5E) or persistence (*E. hydropiper*; Fig 5H and 5I) to those observed in uningested seeds (S1 Appendix; Figs 2 and 5). At the other extreme, 70% of the exotestal seeds of *Allium* entirely lost their seed coat from 30–90% of their surface (Figs 1C, 1D and 3E–3H) (note, however. that ca. 5% of the seeds remained intact, with only cuticle alterations; S1 Appendix). In this latter case, however, even if the seed coat was removed during digestion, the embryos remained shielded within the endosperm with thick (cellulosic) cell walls (Figs 1D and 3G).

Damage to the epidermis (exotesta or exocarp) included the partial removal of epicuticular deposits, either from the more exposed parts of the epidermal cells (e.g., *Lychnis*; Fig 4F–4H) or more or less irregularly from various parts of the seeds in Fabaceae (*Astragalus* and *Glycirrhiza*). Most of the seeds of Fabaceae lost the epicuticular wax from the hilar pads and suffered microfissures, penetrating between the distal ends of the exotestal cells (the "palisade" or "Malpighian cells"; S1 Appendix; Fig 5O, 5P and 5V). In a small proportion of Fabaceae seeds (S1 Appendix), passing caused deeper cracks that pierced through the entire testa or even seed coat (Fig 5P, 5T and 5U). Epidermis cells were entirely destroyed on variable sized portions of the seeds or fruits in *Cuscuta*, *Bolboschoenus* and *Cyperus* (Figs 4Q–4T; 1G, 3L, 3M; 3Q and 3R).

In *Echinochloa cruss-galli*, passing removed the glumes and sterile lemma of spikelets. However, the majority of caryopses (90%) remained enclosed within the fertile lemmas and paleas,

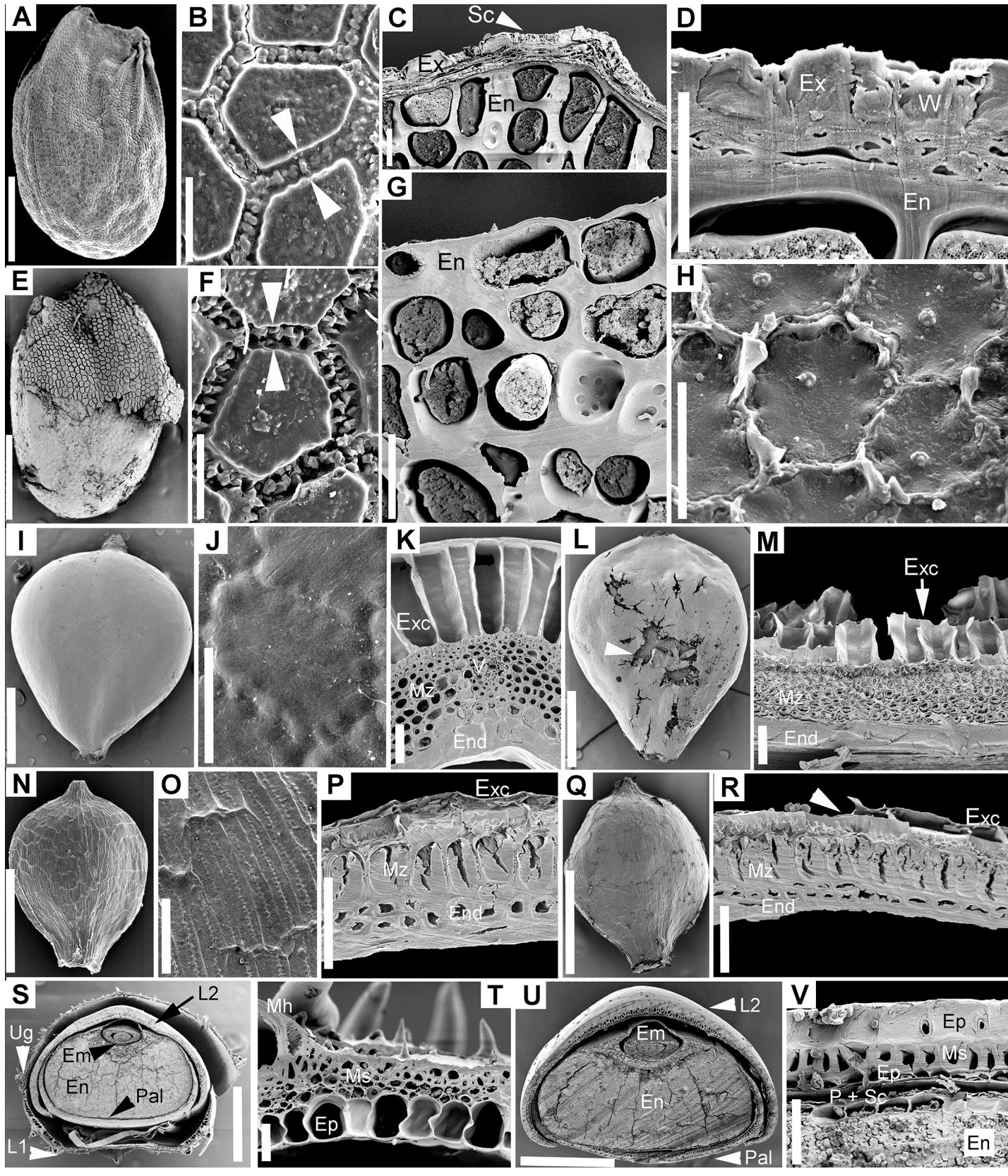

**Fig 3. Scanning electron microscopy of seed coat or pericarp in control and passed diaspores.** A–H. *Allium angulosum.* A–D. Control seeds. A. General morphology. B. Surface of reticulate exotesta; arrows indicate the "zipper-like" epicuticular wax deposits between polygonal exotesta cells. C. Cross-section showing the seed coat and endosperm; note the thick cell walls of endosperm. D. Detail of seed coat showing the layers of exotesta and wax stacks. E–H. Passed seeds. E. Seed coat was entirely removed from portions of the seeds. F. In areas where seed coat persisted, wax stacks between exotesta cells were eroded (arrows). G. Cross-section through area of seed where seed coat was entirely removed and endosperm was left at the exterior of seed. H. Surface of endosperm with remnants of tegmen cell walls. I–M. *Bolboschoenus planiculmis.* I–M. Control achenes. I. General morphology. J. Surface of exocarp. K. Anatomy of pericarp; note the radially elongated cells of the exocarp, which likely also functions as an aerenchyma. L–M. Passed achenes. L. Morphology; arrow indicates damage to pericarp. M. Structure of pericarp in damaged area; note the remnants of exocarp cells. N–R. *Cyperus flavescens.* N–P. Control achenes. N. General morphology. O. Surface of exocarp. P. Cross-section through pericarp. Q. Passed achene. R. Cross-section through pericarp of passed achene (arrow indicates area with exocarp damage). S–V. *Echinochloa crus-galli.* S–T Control spikelet. S. Cross-section showing spikelet layers and lemma/palea surrounding the caryopsis. T. Structure of sterile lemma. U–V. Passed spikelet. U. Cross-section through passed spikelet; note that although the upper glume and sterile lemma were removed, the caryopsis remained intact within the fertile lemma and palea. V. Cross-section through fertile lemma and caryopsis of passed spikelet. Em = embryo; En = endosperm; End = endocarp; Ep = epidermis; Exc = exocarp; Ex = exotesta; Hil = hilum area; L1 = sterile (awned) lemma; L2 = fertile lemma; Mh = macrohair; Mz = mesocarp; Ms = mesophyll; Pal = palea; P = pericarp; Ug = upper glume; V = vascular bundle; W = epicuticular wax. Scale bars = 30 μm except A, E, N, Q, S, U = 0.5 mm; I, L = 1 mm.

which suffered only minor abrasions and entirely preserved their mechanical and chemical integrity (Figs 1N, 1O, 3U and 3V).

Bird feces adhered to the epidermis when the surface of diaspores was prominently reticulated (*Allium* and *Elatine hungarica*) or tuberculate (*Lychnis*).

## 3. Relationships among diaspore traits and between these and survival, germinability and dormancy

Significant correlations showed that larger diaspores generally had thicker seed coats/pericarps and thicker mechanical layers (S1 Table). Based on the extent of damage caused by gut passage (see previous section), five progressive levels of damage were assigned as follows: 1) Minor or no discernable morphological/structural changes compared to controls (*Cirsium, Elatine, Echinochloa*). 2) Changes mostly at the level of the cuticle (e.g., cuticle partially stripped on portions of the seed; *Lychnis*). 3) Fine fissures opened among the distal ends of exotestal cells (*Astragalus, Glychirrhiza*). 4) Epidermis (exotesta or exocarp) was stripped away on portions of the seed/fruit (*Cuscuta, Bolboschoenus, Cyperus*). 5) The entire seed coat was stripped on portions of the seed (*Allium*).

We explored relationships between the damage level and diaspore morpho-anatomical traits, their survival and germinability. All 11 taxa survived gut passage to some extent (Table 2). The highest survival was for *Astragalus contortuplicatus* (51.0%) and *Bolboschoenus planiculmis* (48.7%), whereas the lowest was for *Cirsium brachycephalum* (6.3%) and *Lychnis coronaria* (4.4%). Seed survival was not significantly correlated with diaspore size, morphology or degree of damage (S1 Table). Similarly, seed survival was not significantly different between PD and PY diaspores (t = -0.67, df = 9, P = 0.52).

In all cases, some of the seeds germinated after gut passage (maximum 83%, for *Lychnis coronaria*), although for three of the species none of the control seeds germinated (S3 Table). Only two taxa showed significant effects of gut passage on germinability, with passage increasing germinability in *L. coronaria* but decreasing it in *E. hungarica* (both with PD, S3 Table). The difference between passed and control seeds in germinability for the 11 species showed no significant correlation with diaspore size or morphology, seed survival, degree of damage nor any other diaspore traits (S1 and S2 Tables; $|r| < 0.33$, $|r_s| < 0.1$, $P > 0.05$). Similarly, the difference between passed and control seeds in germinability was not significantly different between PD and PY diaspores (t = -0.06, df = 9, P = 0.96).

PY diaspores were significantly harder (by wet load: t = -3.10, P = 0.013) than PD diaspores, but had significantly lower water permeability (t = -2.62, P = 0.028). No other significant differences between PY and PD were recorded. The degree of damage in passed diaspores was significantly positively correlated with diaspore hardness (wet load: $r_s = 0.63$, $P < 0.05$), but no

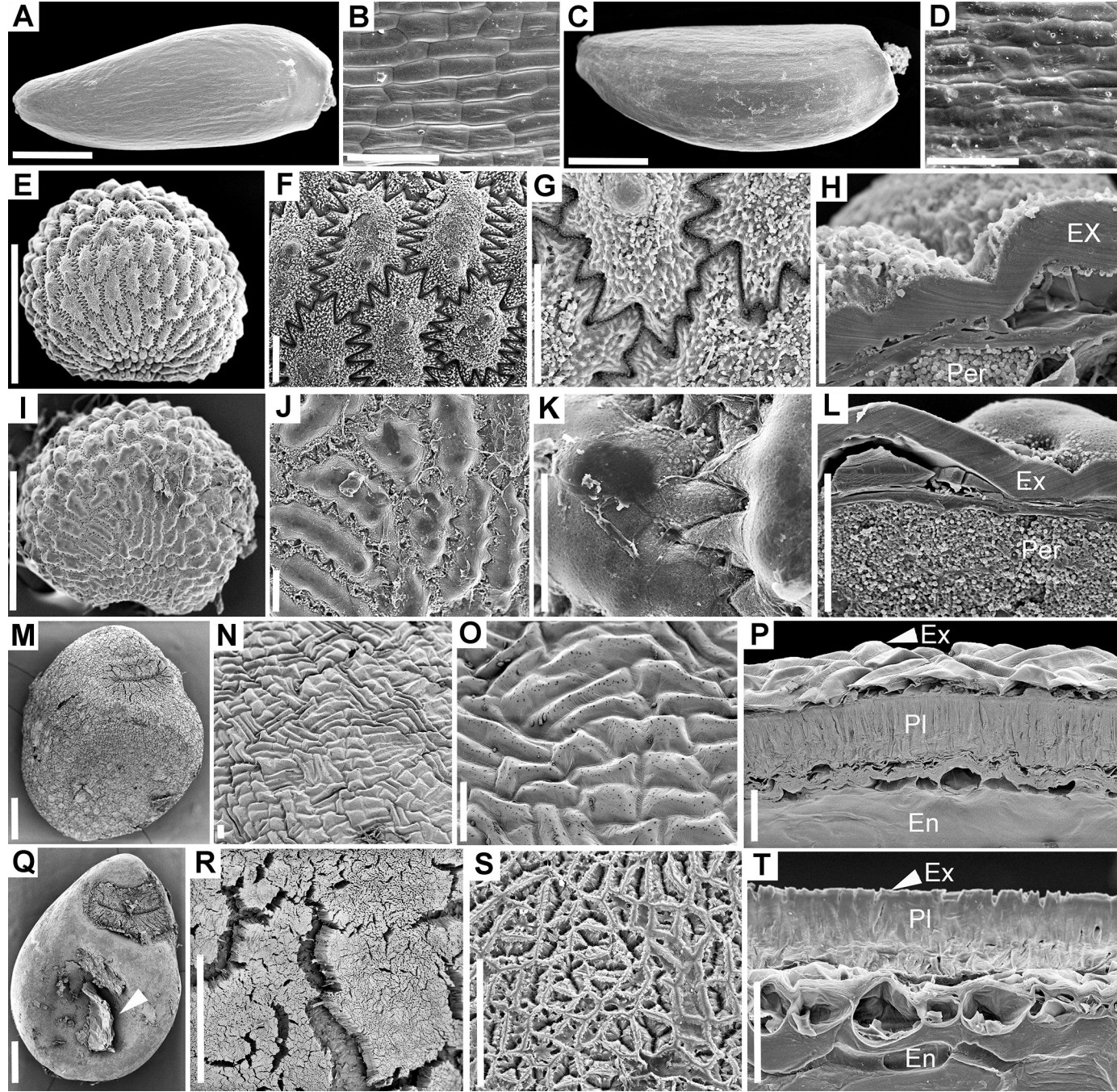

**Fig 4. Scanning electron microscopy of seed coat or pericarp in control and passed diaspores.** A–D. *Cirsium brachycephalum*. A–B. Control cypselae. A. General morphology. B. Surface of exocarp. C–D. Passed cypselae. A. General morphology. D. Surface of exocarp showing minor chips/abrasions of excocarp cuticle. E–L. *Lychnis coronaria*. E–H. Control seeds. E. General morphology. F. Surface of exotesta. G. Detail of exotesta surface. H. Cross-section through the seed coat. I–L. Passed seeds. I. General morphology. J. Surface of exotesta. K. Surface of exotesta detail. L. Cross-section through the seed coat; in all images of passed seeds note that epicuticular wax was stripped from the convex areas of exotesta cells. M–T. *Cuscuta lupuliformis*. M–P. Control seeds. M. General morphology. N–O. Seed surface (exotesta). P. Cross-section through seed coat and endosperm. Q–T. Passed seeds. Q. General morphology; arrow indicates remnants of exotesta after passing. R. Cracks opened within the palisade layer reaching to the endosperm. S. Surface of palisade layer after the removal of exotesta. T. Cross- section through seed coat and endosperm showing the palisade layer left as the most exterior seed layer. Ex = exotesta; Hil = Hilum area; Per = perisperm; Pl = palisade layer. Scale bars. A, C, E, I, M, Q, R = 0.5 mm; B, D, H, L = 30 µm; F, G, J, K, N, O, P, S, T = 50 µm.

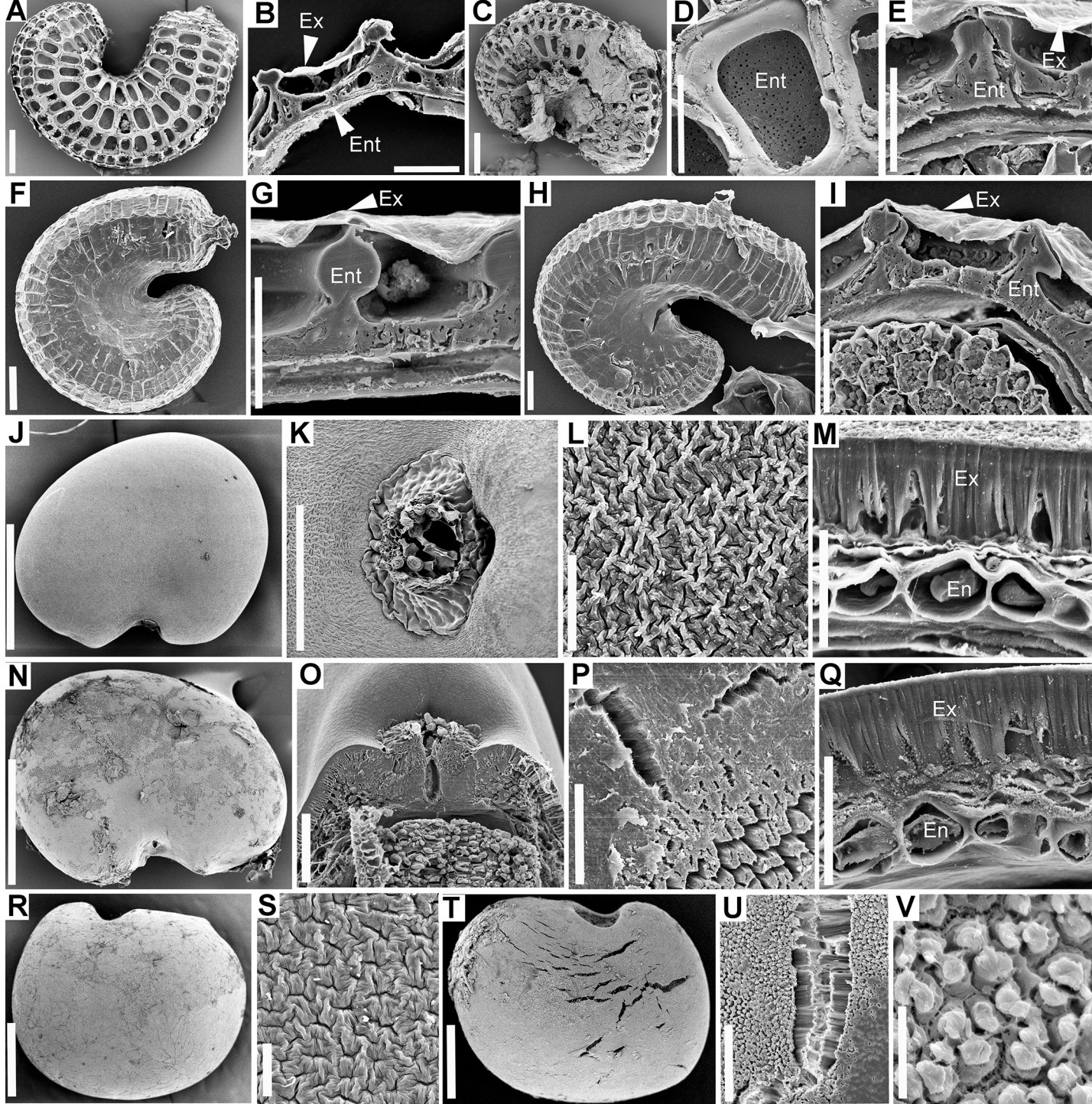

**Fig 5. Scanning electron microscopy of seed coat or pericarp in control and passed seeds.** A–E. *Elatine hungarica*. A–B. Control seeds. A. General morphology. B. Seed coat anatomy. C–E. Passed seeds are virtually unaffected. C. General morphology; note feces adhering to pitted endotesta. D. Surface of endotesta cells showing minute pores in the exterior periclinal cell wall. E. Cross-section through seed coat. F–I. *Elatine hydropiper*. F–H. General morphology of control seeds which are similar to *E. hungarica* but exotesta tends to be persistent in both control and passed seeds. F–G. Control seeds. H–I. Passed seeds. J–Q. *Astragalus contortuplicatus*. J–M. Control seeds. J. Seed morphology. K. Morphology of hilum area. L. Surface of exotesta. M. Cross-section through seed coat. N–Q. Passed seeds. N. General morphology showing removal of cuticle and fine cracks in exotesta. O. Hilum area may present fine cracks in the water gap area. P. Fissures opened among exotestal (Malpighian) cells. Q. Cross-section through seed coat; note the removal of the cuticle. R–V. *Glycyrrhiza echinata*. R–S. Control seeds. R. General morphology. S. Surface of exotesta. T–V. Passed seeds. T. General morphology. U. Detail of crack opened in the exotesta. V. Modification of cuticle in exotesta cells (Malpighian cells). Ent = endotesta; En = endosperm; Ex = exotesta. Scale bars. A, C, F, H, K, O = 100 µm; B, D, E, G, I, M, P, Q, U = 30 µm; L, S, V = 10 µm; J, N = 0.5 mm; R, T = 1 mm.

significant relationships were revealed between the degree of damage and other traits (S2 Table). Hardness (wetload) increased significantly with diaspore size (r = 0.66 for length, r = 0.76 for width, P < 0.05).

## Discussion

### 1. Architecture of the seed coat and pericarp and their defense role: The universality of the mechanical layer(s)

Our statistical analyses suggest that neither diaspore traits *per se*, nor dormancy type are strong predictors of diaspore survival or of the influence of gut passage on germinability. For diaspores that remained "intact" after gut passage, their degree of damage is not a good predictor of germinability. Regardless of the degree of damage, the potential for endozoochory (as indicated by the undamaged embryos, diaspore survival and germination response) is strong for all 11 plant taxa from outside the endozoochory syndrome.

The lack of correlations between the degree of damage and diaspore traits reflects the diversity of diaspore architectures that accompany the universal mechanical layer protection. The seed coat and pericarp are amazingly versatile in their involvement in the protection and dispersal of diaspores. Defense is complex and protection mechanisms unfold at multiple levels—physical, chemical and physiological—to withstand and mediate complex interactions with both abiotic and biotic factors [46]. Large scale comparative studies conducted in angiosperms have shown that all these roles are commonly fulfilled by structurally minimalistic seed coats comprised of only a few layers of cells [47–53]. In contrast to the evolution of ovule integuments which is well known in angiosperms [54, 55], no character evolution study has yet been conducted for seed coat architecture. However, unitegmic ovules, and thus structurally more minimalistic seed coats, have evolved in core-eudicots [48, 54, 56]. More complex seed coats with a fleshy sarcotesta or a huge variety of arils have evolved numerous times in both gymnosperm and angiosperms, and these are typically assumed to be supplementary adaptations for endozoochory or synzoochory (e.g., elaiosomes and myrmecochory; [4, 48, 54, 57, 58].

In the genera or families where the fruit is the diaspore, the seed coat is often reduced and mechanical protection is externally transferred to mechanical cell layers that develop within the pericarp (e.g., the achenes of *Bolboschoenus* and *Cyperus* in this study) or outer covering structures (e.g., the fertile lemmas and paleas of *Echinochloa*; see also [59, 60]). Seed coats that have completely lost their mechanical layer(s) in favour of the pericarp were called "undifferentiated" [48, 45]. Transitions and exceptions are possible (for example in *Cirsium* examined in this study; see more examples in Roth [45].

As also seen in this study, the mechanical protection layer(s) of angiosperm seed coats is often one-celled and lignified, consisting of sclereids, fibers or tracheids; however, thick-cellulosic, silica- or suberin-impregnated cell walls or multiple-celled layers are also possible in various groups of flowering plants [48, 45, 61, 49–53, 62]. Physical protection tissues of dry indehiscent fruits are usually multi-layered sclereids or fibers (see examples from this study, and review by Roth [60]). Exotesta and exocarp cells are cuticularized and generate an incredible array of surface patterns, often with hydrophobic or superhydrophobic properties [63]. Seed coat cells may produce phenolic compounds, anthocyanins, isoflavones and other chemicals, which add a layer of chemical defense to the mechanical protection (e.g., [64]). Internal secretory structures present in the pericarp of dry, indehiscent fruits (e.g., Lamiaceae, Apiaceae, Asteraceae; [60]) secrete a huge variety of compounds with antifungal and antimicrobial effect [e.g., 65, 66]. Mucilage producing seed coats having multiple roles also evolved in numerous plant families (reviewed by [67]).

Diaspores examined in the current study represent a small fraction of the total variation encountered in angiosperms. However, these diaspores from outside the "endozoochory syndrome" proved the effectiveness of structurally simple seed coats and pericarps at protecting diaspores against digestion. The comparative monographs of Corner [48] and Takhtajan [49–53] illustrate the universality and simplicity of mechanical layer(s) present in the seed coat, regardless of the evolution of additional specialized fleshy tissues (see next section). Based on the results of this study, and knowing that other diaspores reported to be dispersed through endozoochory [28, 39, 33, 34, 68, 26, 69] have similar structures, we suggest that the seed coat or pericarp of the vast majority of angiosperm plants possess sufficient mechanical endurance to withstand—with different degrees of success—passing through different types of digestive tracts (e.g., of birds, ungulates or fish). Furthermore, critically there is no major difference in seed structure between plants from an endozoochory syndrome (i.e., with a fleshy fruit) and these other plants dispersed by endozoochory (see below).

## 2. Evolution of protection against endozoochory: Lack of homology for the protective layer(s)

Seeds released from dehiscent fruit or dry-indehiscent fruits form the bulk of diaspores lacking a fleshy fruit yet often dispersed by avian or ungulate endozoochory. Based on phylogenetic evidence, follicle- or capsule-like (dehiscent) fruits are generally accepted to be ancestral, while berries, drupes and dry-indehiscent fruits are considered derived [60, 70–73]. In angiosperm lineages with both dry-dehiscent and fleshy fruits, the latter have evolved from the former numerous times (although cases of reversals are known as well). Examples include Caprifoliaceae, Convolvulaceae, Melastomataceae, Mystaceae, Oleaceae, Rosaceae, Rubiaceae, Solanaceae and many monocots [74–81]. The seed coat data from Corner [48] in conjunction with character evolution results from the above studies suggest that often the evolution of berries does not significantly alter the mechanical architecture of the seed coat. For example, the seed coat of Solanaceae is exotestal with variously lignified epidermal cell walls, regardless of whether their fruit is a capsule (e.g., in *Nicotiana*, *Petunia*) or a berry (e.g., *Atropa*, *Lycium*, *Solanum*, *Withania*; [47, 48, 82, 83]).

Seeds from fleshy fruits and other seeds dispersed by endozoochory often share the same structure of the seed coat within intrafamilial (or family) clades [48–53], and it is the pericarp that undergoes major changes when fleshy fruits evolve [60]. In the case of drupe evolution, the seed coat usually reduces while the pericarp takes over the mechanical protection role, but these changes are similar to those that occur in dry, indehiscent fruit (see below). For example, in Amygaloideae (Rosaceae), drupes are derived from aggregates of follicles or aggregates of capsules [81], and the endocarp ("pit") becomes strongly sclerified [59, 60].

Dry, indehiscent fruits have also evolved multiple times and they figure prominently in monocots (e.g., [81]), Rosales [84], Fagales [85], Polygonaceae [86], and the campanulid families [87]. Although morphologically these fruits often resemble seeds (and they are often called "seeds"), they are only analogous to them, as they originated through additional changes and mechanical trade-offs between the seed coat and the pericarp, not unlike those that occurred in the fleshy fruits. In fact, in a manner comparable to fleshy fruit, dry indehiscent fruits are more specialized, as they often possess additional mechanical layers, as well as additional dispersal-related structures (e.g., those used to assign anemochory or epizoochory syndromes) as, for example, in Dipsacales [88] and Asteraceae [89]. Fleshy fruits evolved sometimes from dry, indehiscent fruits (occasionally with reversals), for example in monocots [80], core Caryophyllales [90] and campanulid families [87]. Such morphologically dramatic diaspore changes within the same plant lineages are usually transitions, reflecting at best degrees of specialisation.

This evolutionary fluidity of form-function in fruits is enabled by genetic changes in key developmental regulatory genes [91]. The gene regulatory networks responsible for the development of "fleshiness", such as MADS-box gene B-sister, have evolved as early as *Ginkgo* [92].

Last but not least, a capacity for endozoochory predates both the angiosperms and gymnosperms, and is not limited to them today. For example, bryophytes and ferns are both dispersed by avian endozoochory [34, 93, 94]. Indeed, protection seems to be the primordial function of the seed covering structures, likely evolved first against abiotic factors (e.g. desiccation, UV protection) and subsequently or concurrently against herbivory (e.g., "fruit is the foliage", [31, 95] and/or invertebrate seed predation. Thus, strictly speaking, both in the case of fleshy and "unspecialised" diaspores, the protective layers are probably an exaptation (sensu [96]) for endozoochory.

## 3. Reassessing the value of the endozoochory syndrome

Too often, endozoochory has been equated with frugivory and the presence of a fleshy-fruit, hence, less attention has been paid to endozoochory of other angiosperms, e.g. by granivorous waterbirds, which is a widespread mechanism of importance for many plants, including those assigned to "unassisted dispersal" (see Introduction). The use of functional traits in ecology is booming and trait databases are providing the basis for much research, but models based on these databases cannot be expected to be reliable whilst endozoochory is ignored for the flora which lacks a fleshy fruit (i.e., whilst models attempt to predict dispersal based purely on syndromes). For a few examples of the many studies predicting plant dispersal while ignoring non-classical endozoochory, see [97–99].

By comparing the morphology and anatomy of diaspores of a broad taxonomic range of taxa before and after gut passage by granivorous waterfowl, we have illustrated how resistance to gut passage is widespread, and involves the ubiquitous presence of mechanical layers as well as a range of different diaspore traits. Van der Pijl [4] created a category of "non-adapted" diaspores to cover the case of fruits/seeds dispersed through endozoochory but which lack the attractively colored, fleshy edible part. However, our findings suggest these diaspores may be no less "adapted" to endozoochory than those with an "endozoochory syndrome." Our study suggests that the evolution of "external flesh" represents a specialisation, but nevertheless just one of the possible diaspore architectures evolved as a result of mutualistic interactions. Granivory can also provide dispersal benefits to the plants whose diaspores are ingested, and hence be the basis for seed-dispersal mutualisms in a comparable manner to frugivory. The key difference is that the plant pays a cost as a fraction of seeds ingested instead of the cost of the fleshy-fruit. Yet, unfortunately the existence of granivory mutualisms is often ignored in the literature about dispersal mutualisms (e.g., [100]).

A major research effort is required to establish the importance of non-classical endozoochory in different ecosystems, especially for the high proportion of angiosperms typically assigned to the "unspecialized syndrome" (including the majority of the European flora; [17]. Major networks of mutualistic dispersal interactions between granivores or herbivores and plants are poorly understood, but such ecological connections are essential for biodiversity conservation [101]. Dispersal interactions are also major determinants of the species composition and gene flow in biological communities, and a vital part of the architecture of biodiversity, or "interactome" [102]. Non-classical endozoochory is well suited to long-distance dispersal and the maximum dispersal distances expected from migratory waterbirds greatly exceed those that can be expected from abiotic vectors [103–104]. Maximum dispersal distances are central to predicting changes in plant distribution in response to climate change, or biological invasions [97–98].

Further work is also required to establish which diaspore traits are useful predictors of non-classical endozoochory. Existing work on ungulates and waterfowl suggest that a combination of relatively small size, greater hardness and roundness favours seed survival during gut passage [33, 68, 105]. We hope that our study will promote a growth in research into non-classical endozoochory.

## Supporting information

**S1 Appendix. Comparative morphology and anatomy of seed coat/pericarp in control and passed diaspores.**
(DOCX)

**S2 Appendix. Glossary of seed/fruit terms used in the article.**
(DOCX)

**S1 Table. Pearson correlation matrix for diaspore traits.**
(DOCX)

**S2 Table. Spearman non-parametric correlation matrix for diaspore traits including the degree of damage for passed seeds in five categories (degree of damage).**
(DOCX)

**S3 Table. Results of germination tests and effect of gut passage on germinability tested with fisher's exact test.**
(DOCX)

## Acknowledgments

We thank Alexander Sukhorukov and one anonymous reviewer for their suggestions.

## Author Contributions

**Conceptualization:** Mihai Costea, Andy J. Green.

**Data curation:** Mihai Costea, Hiba El Miari, Levente Laczkó, Réka Fekete, Ádám Lovas-Kiss.

**Formal analysis:** Mihai Costea, Andy J. Green.

**Investigation:** Mihai Costea, Ádám Lovas-Kiss.

**Methodology:** Mihai Costea, Ádám Lovas-Kiss.

**Supervision:** Attila V. Molnár.

**Visualization:** Mihai Costea, Hiba El Miari.

**Writing – original draft:** Mihai Costea, Andy J. Green.

**Writing – review & editing:** Mihai Costea, Andy J. Green.

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
