## [Decision Letter · Decision Letter 0]

13 Nov 2019

PONE-D-19-28800

The Effect of Gut Passage by Waterbirds on the Seed Coat and Pericarp of Diaspores Lacking “External Flesh”: Evidence for Widespread Adaptation to Endozoochory in Angiosperms

PLOS ONE

Dear Dr. Mihai Costea,

Thank you for submitting your manuscript to PLOS ONE. After careful consideration, we feel that it has merit but does not fully meet PLOS ONE’s publication criteria as it currently stands. Therefore, we invite you to submit a revised version of the manuscript that addresses the points raised during the review process.

The reviewers coincide that the manuscript is interesting and well presented. However, there are some minors concerns that the authors should address. 

We would appreciate receiving your revised manuscript by December 12th. To enhance the reproducibility of your results, we recommend that if applicable you deposit your laboratory protocols in protocols.io, where a protocol can be assigned its own identifier (DOI) such that it can be cited independently in the future. For instructions see: http://journals.plos.org/plosone/s/submission-guidelines#loc-laboratory-protocols

We look forward to receiving your revised manuscript.

Kind regards,

Guillermo C. Amico

Academic Editor

PLOS ONE

Journal Requirements:

Additional Editor Comments:

The reviewers coincide that the manuscript is interesting and well presented. However, there are some minors concerns that the authors should address.

Reviewers' comments:

Reviewer's Responses to Questions

**Comments to the Author**

1. Is the manuscript technically sound, and do the data support the conclusions?

Reviewer #1: Yes

Reviewer #2: Yes

2. Has the statistical analysis been performed appropriately and rigorously? 

Reviewer #1: Yes

Reviewer #2: Yes

3. Have the authors made all data underlying the findings in their manuscript fully available?

Reviewer #1: Yes

Reviewer #2: Yes

4. Is the manuscript presented in an intelligible fashion and written in standard English?

Reviewer #1: Yes

Reviewer #2: Yes

5. Review Comments to the Author

Reviewer #1: Dear all,

The paper is interesting and is suitable for publication. One major question is arisen: are you really sure that the "Ex" in Cirsium is really seed coat. We have studied some Asteraceae (published or in review) and recognized that the sclerenchyma belongs to the pericarp , and the seed coat is represented by two or several parenchymatous cell layers. Based on that, I ask you to check whether the exotestal layer of the seed coat is really a part of seed coat and not a part of pericarp.

Best wishes,

Reviewer #2: The experiment was well designed and the data collected and analyzed with sound technical methods. The data supported the conclusions and summaries. A well written paper!

Minor revision suggested in:

1. have the controlled seeds going through the same preparation treatment? i.e., disinfection treatment ( 1 minutes in 25% NaOCL). This not clearly stated in the section "2. Germination tests".

2. provide a brief justification for the same germination conditions (i.e., light period and temperature alternation) for all 11 species, in the section "2. Germination tests".

6. PLOS authors have the option to publish the peer review history of their article (what does this mean?). If published, this will include your full peer review and any attached files.

Reviewer #1: Yes: Alexander P. Sukhorukov

Reviewer #2: No

---

## [Author Response · Author response to Decision Letter 0]

26 Nov 2019

Editor

1) Checked the funding sources and corrected accordingly. Also, deleted this funding info from the Acknowledgments.

2) Added a column with geographical location of collecting sites in Table 1 and permit numbers for collecting rare species (mentioned) from non-protected public area in the "material and methods" section (lines 149-152).

Reviewer #1: The paper is interesting and is suitable for publication. One major question is arisen: are you really sure that the "Ex" in Cirsium is really seed coat. We have studied some Asteraceae (published or in review) and recognized that the sclerenchyma belongs to the pericarp, and the seed coat is represented by two or several parenchymatous cell layers. Based on that, I ask you to check whether the exotestal layer of the seed coat is really a part of seed coat and not a part of pericarp.

Authors: We have verified again the identity of the mechanical layer in Cirsium (Carduoideae, Asteraceae) and confirm that this cell layer is indeed the seed coat exotesta, and that it does not belong to the pericarp. The reviewer, Dr. Alexander Sukhorukov, studied Tragopogon and more generally Scorzonerinae in Asteraceae, which have a different organization of the seed coat/pericarp architecture. In these latter groups, the defense layers has indeed been transferred from the seed coat to the pericarp. The following references, among others, confirm our decision:

Häffner, E., 2000. On the phylogeny of the subtribe Carduinae (tribe Cardueae, Compositae). Englera, pp.: 3-208.

Ozcan, M., 2017. Cypsela micromorphology and anatomy in Cirsium sect. Epitrachys (Asteraceae, Carduoideae) and its taxonomic implications. Nordic Journal of Botany, 35: 653-668.

Ozcan M. 2019. Carpological investigations on some Cirsium (Asteraceae, Carduoideae) taxa from NE Anatolia. Acta Botanica Hungarica. 61: 369-385.

Roth, I. 1977. Fruits of Angiosperms; Gebruder Borntraeger, Berlin, Stuttgart; pp.: 258-270.

Zarembo, E.V. and Boyko, E.V. 2008. Carpology of some East Asian Cardueae (Asteraceae). Anales del Jardín Botánico de Madrid. 65: 129-134.

Reviewer #2: The experiment was well designed and the data collected and analyzed with sound technical methods. The data supported the conclusions and summaries. A well written paper!

Minor revision suggested in:

1. Have the controlled seeds going through the same preparation treatment? i.e., disinfection treatment ( 1 minutes in 25% NaOCL). This not clearly stated in the section "2. Germination tests".

Authors: we made an addition (line 172) to indicate that both control and treated diaspores were treated the same way.

2. Provide a brief justification for the same germination conditions (i.e., light period and temperature alternation) for all 11 species, in the section "2. Germination tests".

Authors: That would not be necessary because as indicated in the text, the objective was to determine if the seeds can germinate, not to determine their specific germination requirements.

---

## [Editor Report · Decision Letter 1]

3 Dec 2019

The Effect of Gut Passage by Waterbirds on the Seed Coat and Pericarp of Diaspores Lacking “External Flesh”: Evidence for Widespread Adaptation to Endozoochory in Angiosperms

PONE-D-19-28800R1

Dear Dr. Mihai Costea,

We are pleased to inform you that your manuscript has been judged scientifically suitable for publication and will be formally accepted for publication once it complies with all outstanding technical requirements.

With kind regards,

Guillermo C. Amico

Academic Editor

PLOS ONE

Additional Editor Comments (optional):

Thank you for your latest revisions on your manuscript for PLOS ONE. I found that you responded completely to the previous reviewer's comments and I did not find any further concerns. I have enjoyed reading your manuscript and it will have a substantial impact on this field.

---

## [Editor Report · Acceptance letter]

5 Dec 2019

PONE-D-19-28800R1 

The Effect of Gut Passage by Waterbirds on the Seed Coat and Pericarp of Diaspores Lacking “External Flesh”: Evidence for Widespread Adaptation to Endozoochory in Angiosperms 

Dear Dr. Costea:

I am pleased to inform you that your manuscript has been deemed suitable for publication in PLOS ONE. Congratulations! Your manuscript is now with our production department. 

With kind regards,

on behalf of

Dr. Guillermo C. Amico 

Academic Editor

PLOS ONE